# Evaluating Steering Techniques using Human Similarity Judgments

## Abstract

Current evaluations of Large Language Model (LLM) steering techniques focus on task-specific performance, overlooking how well steered representations align with human cognition. Using a well-established triadic similarity judgment task, we assessed steered LLMs on their ability to flexibly judge similarity between concepts based on size or kind, two central dimensions organizing human mental representations. We found that prompt-based steering methods outperformed other methods both in terms of steering accuracy and model-to-human alignment. We also found LLMs were biased towards 'kind' similarity and struggled with 'size' alignment. This evaluation approach, grounded in human cognition, adds further support to the efficacy of prompt-based steering and reveals privileged representational axes in LLMs prior to steering.

## 1 Introduction

Central to the flexibility of human cognition is the ability to marshal both one's knowledge of the world and the current context to guide behavior (Wong et al., 2025). Different aspects of a learned concept can be preferentially activated to execute different tasks. For example, the *size* of an orange might be important when a shopper is deciding how many can fit inside their shopping basket, whereas the *kind* of produce it is (i.e., fruit) might be important when a grocer is arranging their wares on a shelf. Such flexible behavior is facilitated by learning robust representations of concepts, captured by theories and models of semantic knowledge (Rogers & McClelland, 2004; Rogers, 2024; Saxe et al., 2019), and by 'guiding' these representations through context-sensitive mechanisms, captured by accounts of cognitive/semantic control (Cohen et al., 1990; Miller & Cohen, 2001; Ralph et al., 2017; Giallanza et al., 2024).

These aspects of human semantic cognition—representation and control—provide strong analogies to two important axes of evaluation for large language models (LLMs): (1) representation learning through pre-training and (2) intervention-based steering. This isomorphism is important insofar as cognitive scientists have developed methods for characterizing human mental representations under varied contexts for naturalistic tasks that underpin a variety of human behaviors. Here, we focus on evaluating current LLM steering techniques through the lens of cognitive science-inspired methods, which constitute a critical complement to efforts in interpretability research, primarily stemming from the computer sciences. This allows for evaluating steering techniques not just in terms of performance, but in terms of how aligned the resultant model behaviors and representations are to those of humans when presented with qualitatively similar interventions.

In the present work, we used a triadic similarity judgment task (Sievert et al., 2023; Hebart et al., 2020), a technique that has been shown to be effective at characterizing human mental representations, where humans judged the similarity between concepts in terms of either **size** or **kind**. We then tested a suite of LLM intervention steering techniques on `gemma2-27b` and `gemma2-9b` and assessed each method's (1) *accuracy*, measured as the number of 'correct' judgments on the size and kind tasks, and (2) *human alignment*, measured as the Procrustes correlation between embeddings generated from human judgments versus those generated from model judgments. Model accuracy differed dramatically across steering methods, with some approaches yielding human-level performance; but surprisingly, human alignment was poor, especially for size judgments across all steering methods. These results suggest critical differences between semantic control in humans and large language models.

## 2 RELATED WORK

### 2.1 TRIADIC JUDGMENT TASKS

Triadic judgment tasks, coupled with advances in embedding algorithms (Jamieson et al., 2015; Sievert et al., 2023; Muttenthaler et al., 2022) have enabled the characterization of the representational geometry underlying human concepts (Jamieson et al., 2015; Sievert et al., 2023; Muttenthaler et al., 2022; Hebart et al., 2020; Muttenthaler et al., 2023b; Suresh et al., 2024; Giallanza et al., 2024). The general approach is to present participants with three concepts and to ask them to indicate either the odd one out or which of two options is most similar to a designated target (Sievert et al., 2023; Muttenthaler et al., 2022). These judgments are used to estimate a low-dimensional embedding where similar concepts are closer together. These techniques have been extended to AI systems, specifically vision models (Muttenthaler et al., 2023a; Mukherjee & Rogers, 2025), to estimate how human-like neural network representations are (see Sucholutsky et al., 2023). Judgments made by LLMs on such tasks can also be used to estimate model embeddings, helping uncover representational structures in otherwise opaque systems (Hebart et al., 2020; Sucholutsky et al., 2023). This approach has revealed fundamental differences: human conceptual structures remain consistent across cultures, while LLMs exhibit task-dependent variation (Suresh et al., 2023). Applications extend to standardized similarity norms (Hout et al., 2022) and semantic organization principles (Mirman et al., 2017).

### 2.2 MODEL STEERING METHODS

Although language models are already context-sensitive due to their outputs being conditioned on prior text, steering methods further control behavior via internal interventions or structured prompts. Because these methods allow for fine-grained control over LLM behavior at inference time, they offer a potential alternative to more expensive methods–such as supervised fine tuning (SFT)–which directly modify model weights (Wu et al., 2025).

**Prompting**  In many instances, model outputs can be steered directly through prompt instructions provided in natural language (Sahoo et al., 2025). *Zero-shot prompting* involves providing a single task instruction or example to direct LLM outputs, and is capable of shaping responses to support emergent behaviors including multi-step reasoning (Kojima et al., 2023), social simulation (Chuang et al., 2024) and image classification (Abdelhamed et al., 2025). Even without explicit instruction, LLMs can induce general rules and abstractions from minimal examples via *in-context learning*. This is sufficient to steer model behavior for translation tasks, (Brown et al., 2020), logical reasoning (Yin et al., 2024), and task induction (Honovich et al., 2022).

**Task vectors**  Hendel et al. (2023) find that LLMs encode in-context task instructions as a linear direction in intermediate layers. This direction, $\theta$, is a compressed representation of the abstract rule underlying the sequence. When $\theta$ is extracted from the forward pass on a rule-based prompt $x$ and patched into the forward pass of an empty prompt $x'$ with no task instructions, the model is able to produce the correct sequence completion for the empty prompt. Initial work by Hendel et al. (2023) demonstrates the efficacy of task vectors for steering model behavior in domains including translation, semantic knowledge (country-capitol mappings) and syntactic rules. Todd et al. (2024) extends this general paradigm to recover finer-grained representations from the activations of individual attention heads. Subsequent work has since applied task vectors to causal interventions in a variety of applications from visual image-masking tasks (Hojel et al., 2024) to more naturalistic domains (Kang et al., 2025), while improvements to the paradigm itself include training mechanism modifications to induce task vectors (Yang et al., 2025) and the superposition of multiple task vectors during in-context learning (Xiong et al., 2024).

**Difference-in-means (DiffMean)**  While task vectors demonstrate, in-practice, that transformer hidden layers encode abstract rules for task representations, these representations are often noisy as a result of extracting the *entire* hidden state at a given token position (Hendel et al., 2023). Marks & Tegmark (2024) introduce a fine-grained method for extracting the steering vector as a linear direction $\theta$ in the activation space that represents the difference between the mean hidden states of positive and negative prompt examples. This vector, represented as $\theta$, consistently shifts model

activations along the direction $\theta$, from positive to negative or negative to positive (where *negative* and *positive* are examples along some dimension). Marks & Tegmark (2024) demonstrate that these vectors capture significantly less noise compared to simple linear probes, and generalize to a range of naturalistic domains.

**Sparse autoencoders**    A key difficulty in extracting steering directions from model activations is the dense nature of these activations: a single neuron will activate for many different tasks and concepts as a part of the distributions representing those concepts. *Sparse autoencoders* attempt to solve this issue by learning a compressed set of activations $\mathbf{z}$ corresponding to the activations in a given transformer layer $l$, with a sparsity penalty applied $\mathcal{L}_{\mathrm{sparse}}$ (Cunningham et al., 2023). This allows for the unsupervised discovery of human-interpretable concepts, represented by the activations of single SAE neurons. Formally, the SAE objective is

$$\mathcal{L} = \|\mathbf{h}^{(l)} - D(E(\mathbf{h}^{(l)}))\|_2^2 + \lambda\,\mathcal{L}_{\mathrm{sparse}},$$

where $\mathbf{h}^{(l)}$ are the transformer activations, $E$ and $D$ are the encoder and decoder, and $\lambda$ is the sparsity penalty. The concepts identified by SAEs range from world knowledge and semantic properties to syntactic features and compositional abstractions (Shu et al., 2025). Interestingly, SAE neurons can be used to steer model outputs along dimensions of interest, such as translating LLM outputs to French given the activation of a corresponding latent feature in the sparse autoencoder (He et al., 2025). While the extent to which SAEs provide a practical means of model steering and interpretability is disputed (Wu et al., 2025), subsequent work provides methods for obtaining more interpretable, coherent concepts (Bussmann et al., 2025; Rajamanoharan et al., 2024).

### 2.3    PERFORMANCE VERSUS ALIGNMENT

Task performance and representational alignment are distinct axes of evaluation. Linsley et al. (2023) found that higher object recognition accuracy led to poorer alignment with human visual cortex. Liu et al. (2023) reported tradeoffs between alignment and task performance. This suggests that models can match human performance but rely on different internal representations (Piantadosi & Hill, 2021). As De Bruin et al. (2024) argue, evaluations should assess not only what systems do, but how. Thus, we evaluate both competence and alignment.

## 3    METHODS

### 3.1    DATASET

We use the Round Things Dataset (Giallanza et al., 2024) to evaluate these aspects. This dataset comprises 46 concrete objects: 25 human-made artifacts and 21 natural kinds (fruits and vegetables). All items are roughly spherical in shape, varying along a discrete *kind* dimension (artifacts versus plants) and a continuous *size* dimension. The clear, unambiguous ground truth nature of the underlying dimensions governing size and kind, coupled with the fact that these are dimensions people consider in everyday human life make this a uniquely well scoped yet ecologically relevant dataset for assessing the effects of steering in language models. We describe the dataset in further detail below.

**Dataset design.**    The Round Things Dataset was specifically designed to investigate how semantic representations can be contextually modulated along orthogonal dimensions. The choice of roughly spherical objects serves multiple methodological purposes: (1) it facilitates size comparisons by minimizing shape-based confounds, (2) it ensures that size judgments are based primarily on scale rather than geometric complexity, and (3) it creates a controlled stimulus set where the two key dimensions of interest—kind and size—are orthogonal rather than confounded.

The dataset was constructed to include commonly known objects to ensure high familiarity across participants. Each item possesses an empirically-derived ground-truth average diameter obtained through systematic internet searches. Specifically, the authors conducted Google searches using the phrase "average diameter of a ____," recording the answer yielded by search engine summaries. When searches returned size ranges, the midpoint was taken as the median diameter. For items that naturally occur in multiple sizes (e.g., pumpkins, yoga balls), searches were refined by adding "medium-size" to obtain more standardized measurements.

**Data composition.** The 46 items are distributed across two semantic categories: 25 human-made artifacts (such as balls, spherical tools, and round household objects) and 21 fruits and vegetables (such as apples, oranges, and other roughly spherical produce). Importantly, object sizes are approximately balanced across the two categories, ensuring that neither semantic domain is systematically associated with larger or smaller items. This balanced distribution is crucial for investigating how semantic category membership interacts with size-based similarity judgments without confounding effects.

The dataset thus provides stimuli that differ along two clear but unrelated abstract semantic dimensions: the discrete taxonomic dimension of kind (artifact versus natural) and the continuous physical dimension of size. This orthogonal structure allows for controlled investigation of how task context can selectively emphasize one dimension while de-emphasizing another, and whether such emphasis preserves or eliminates information from the non-target dimension.

Human behavioral data collected using this dataset has been used to derive two-dimensional embeddings via crowd kernel ordinal embedding algorithms, allowing for quantitative analysis of how different task contexts reshape the geometric organization of semantic space while preserving cross-context similarities.

## 3.2 TRIADIC SIMILARITY JUDGMENT TASK

We employ a triadic judgment task where both humans and LLMs identify which of two items $x_1$ or $x_2$ is most similar to a reference item $x_{\text{ref}}$ along a specified semantic dimension. Participants are instructed to judge which of the two options is most similar to the target item either "in terms of size" or as "a more similar kind of thing."

This paradigm is motivated by the hypothesis that when people estimate similarity along specific dimensions, they use task instructions to steer their semantic representations toward task-relevant features. We test whether different LLM steering methods can similarly modulate representations to achieve both high accuracy (competence) and human-like similarity patterns (alignment).

The dataset's balanced design—with sizes approximately distributed across both semantic categories—enables investigation of how task instructions influence the relative weighting of size versus kind information in similarity computations. This provides insights into whether context-dependent steering preserves or eliminates information from non-target dimensions, revealing the flexibility and constraints of semantic representations under controlled processing conditions.

## 3.3 STEERING METHODS

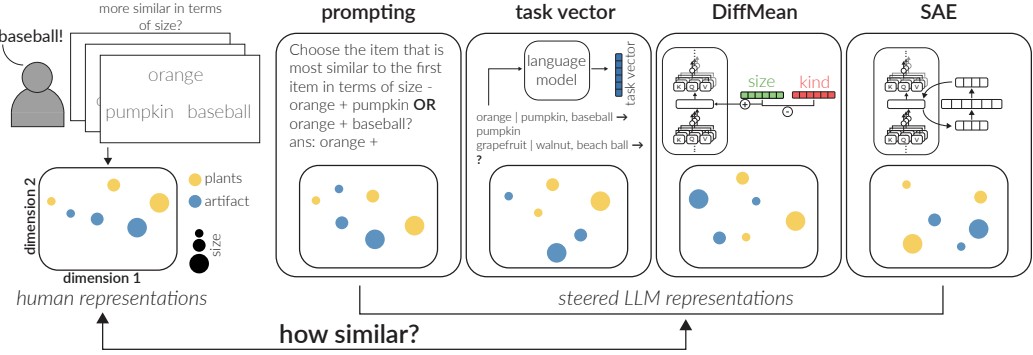

Figure 1: General workflow for deriving human and steered model embeddings.

We evaluated four different steering methods—prompting, task vectors, DiffMean, and SAEs— alongside two prompting baselines (zero-shot and in-context) for comparison. All methods were tested on two open-source LLMs capable of running on a single NVIDIA H100 GPU: `gemma2-27b` and `gemma2-9b`. Figure 4 shows the general workflow for applying these steering methods to the triadic judgment task.

For the prompting-based approaches, different steering conditions required tailored prompts to elicit appropriate similarity judgments along specific semantic dimensions. Each prompt was designed to direct the model's attention toward either kind-based or size-based similarity, or to allow for general similarity assessment in the neutral condition, as shown in Table 1. All trials received independent prompt evaluation to ensure consistent application of each steering method.

For the activation-based steering methods (task vectors, DiffMean, and SAEs), we applied the corresponding interventions to the model's internal representations while using a consistent neutral prompt format across all trials. This approach allows us to isolate the effects of representational steering from prompt-based steering, providing a comprehensive comparison of different steering mechanisms.

Detailed implementation specifics for each steering method are provided in Appendix A.1.

Table 1: Prompt templates used for different steering conditions. The placeholder {p} is replaced with the triplet comparison question: "Which item is most similar to {item_x}: {item_y} or {item_z}?"

| Condition | Prompt Template |
|---|---|
| Kind | Choose the second item that is most similar to the first item in terms of the KIND of thing it is. Respond only with the name of the item exactly as written. {p} |
| Size | Choose the second item that is most similar to the first item in terms of SIZE. Respond only with the name of the item exactly as written. {p} |
| Neutral | Choose the second item that is most similar to the first item. Respond only with the name of the item exactly as written. {p} |

## 3.4 ANALYSIS APPROACH

**Embedding algorithm.** We applied an ordinal embedding algorithm (Tamuz et al., 2011; Sievert et al., 2023) to similarity judgments from both humans and models, creating semantic spaces where frequently co-judged similar items were positioned closer together. The embedding construction process minimized crowd-kernel triplet loss (Tamuz et al., 2011) by optimizing Euclidean distances between word pairs. To ensure reliable estimates, we reserved 20% of our data for validation purposes and monitored crowd-kernel loss throughout the fitting procedure.

Under this framework, each triplet judgment yields an ordinal constraint of the form "item $i$ is closer to item $j$ than to item $k$." Let $D^\star$ be the squared Euclidean distance matrix of unknown embedding points $\{x_i^\star\}_{i=1}^n \subset \mathbb{R}^d$. The ordinal embedding model assumes a monotone link function (e.g., logistic):

$$\Pr\left[y_{(i;j,k)} = 1\right] = f\left(D_{ik}^\star - D_{ij}^\star\right), \qquad f(0) = \tfrac{1}{2}. \tag{1}$$

The ordinal embedding algorithm minimizes an empirical surrogate of the negative log-likelihood using crowd-kernel triplet loss over a low-rank parameterization.

*Sample complexity considerations.* If the true embedding has rank $d$ and triplet judgments are sampled approximately uniformly at random, then with high probability the out-of-sample prediction error is

$$\mathcal{E}_{\text{pred}} = \tilde{O}\left(\sqrt{\frac{d\,n\log n}{|S|}}\right), \qquad \Rightarrow \quad |S| = \tilde{\Theta}(d\,n\log n) \tag{2}$$

Thus, $\tilde{\Theta}(nd\log n)$ triplet judgments suffice for accurate prediction of new comparisons, and at least $\Omega(d\,n\log n)$ ordinal comparisons are information-theoretically necessary (Jain et al., 2016). For our implementation with $n = 46$ items and $d = 2$ dimensions, we collected $N_{\text{triplets}} = c \cdot d \cdot n\log n$ triplet judgments per steering method (where $c \approx 5$ based on our 2,500 judgments), balancing statistical efficiency with computational constraints.

**Accuracy measurement.** We evaluated the competence of each steering method by measuring LLM accuracy on the triadic judgment task compared to ground-truth and human performance. Human accuracy benchmarks were derived from human embeddings by evaluating the same 2,400 triplets used in model assessments: when the Euclidean distance from $x_{\text{ref}}$ to $x_1$ was shorter than from $x_{\text{ref}}$ to $x_2$, we classified $x_1$ as the correctly identified more similar item. This approach enables direct comparison between human and model performance on identical triplet sets, providing a unified measure of task competence across all steering conditions.

**Alignment analysis.** To quantify alignment between human and LLM representations, our main measure of human-model alignment was the amount of variance in human semantic embeddings explained by model-based semantic embeddings ($R^2$) after Procrustes aligning the two spaces (**?**). Procrustes transformation aligns two vector spaces of equal dimensionality by finding the optimal set of linear transformations (rotations, scalings, and translations) to minimize the sum of squared distances between corresponding points in the two spaces (residual SSE). Using the sum of squared distances in the human embeddings (human SSE) as the target, we computed an $R^2$ metric as:

$$R^2 = 1 - \frac{\text{residual SSE}}{\text{human SSE}}$$

This approach measures how well variations in pairwise distances in LLM-derived representations correspond to those in human representations after allowing for optimal linear transformations in the representational geometry. Since both embedding spaces are of the same dimensionality and permitting these linear transformations makes our alignment estimate maximally generous, this provides a robust upper bound on the correspondence between human and model semantic representations.

## 4 RESULTS

We assessed whether steering led to *correct judgments* (competence) and how the steering influenced the underlying *representational geometry* in models with respect to that of humans (alignment).

### 4.1 MODEL REPRESENTATIONS SHOW DEFAULT KIND ALIGNMENT

Figure 3 shows how successful different steering methods were in guiding LLMs to make accurate kind and size judgments. Neutral prompts that simply asked LLMs to indicate which of the two options was most similar to the target, without additional context, were better aligned to kind judgments than to size judgments. This is evidenced by the significantly higher accuracy of model kind judgments in predicting neutral judgments relative to size ($\beta_{\text{kind}} = 2.86$, $p < 0.001$, $\beta_{\text{size}} = 0.01$, $p = 0.89$). We observed a similar pattern in the alignment of neutral model judgments to *human* kind representations. (bottom row of Figure 3): Neutral prompt embeddings were strongly aligned with human kind embeddings but showed no alignment with human size embeddings. For the kind dimension, both 27B and 9B variants correlated highly with human judgments ($r_{\text{27B,kind}} = 0.724$, $p = 2.52 \times 10^{-4}$; $r_{\text{9B,kind}} = 0.708$, $p = 4.06 \times 10^{-4}$). In contrast, correlations with human size embeddings were near zero and nonsignificant ($r_{\text{27B,size}} = 0.11$, $p = 0.659$; $r_{\text{9B,size}} = 0.128$, $p = 0.607$). All $p$-values for alignment hypothesis tests were computed using a Fisher $r \rightarrow z$ transform.

### 4.2 PROMPTING OUTPERFORMS OTHER STEERING METHODS AT PREDICTING GROUND TRUTH AND HUMAN ALIGNMENT

Intervention methods (SAEs, task vectors, and DiffMean) consistently performed worse than prompting in producing correct (ground-truth) judgments for both the *kind* ($\beta = 0.16$, $t = 15.76$, $p < 0.001$) and *size* conditions ($\beta = 0.30$, $t = 26.68$, $p < 0.001$). In the *kind* condition, task vectors and DiffMean performed equivalently in their accuracy profiles (as reflected in the small residual differences captured by the regression model), while in the *size* condition we observed a small but reliable benefit in performance for task vectors over DiffMean ($\chi^2 = 8.79$, $p = 0.003$). SAEs were more effective at predicting the ground truth in the *size* condition relative to DiffMean and task vectors ($\chi^2 = 28.25 - 47.02$, $p < 0.001$), but were consistently worse in the *kind* condition relative to other steering methods ($\chi^2 = 1.85 - 3.23$, $p = 0.07 - 0.17$).

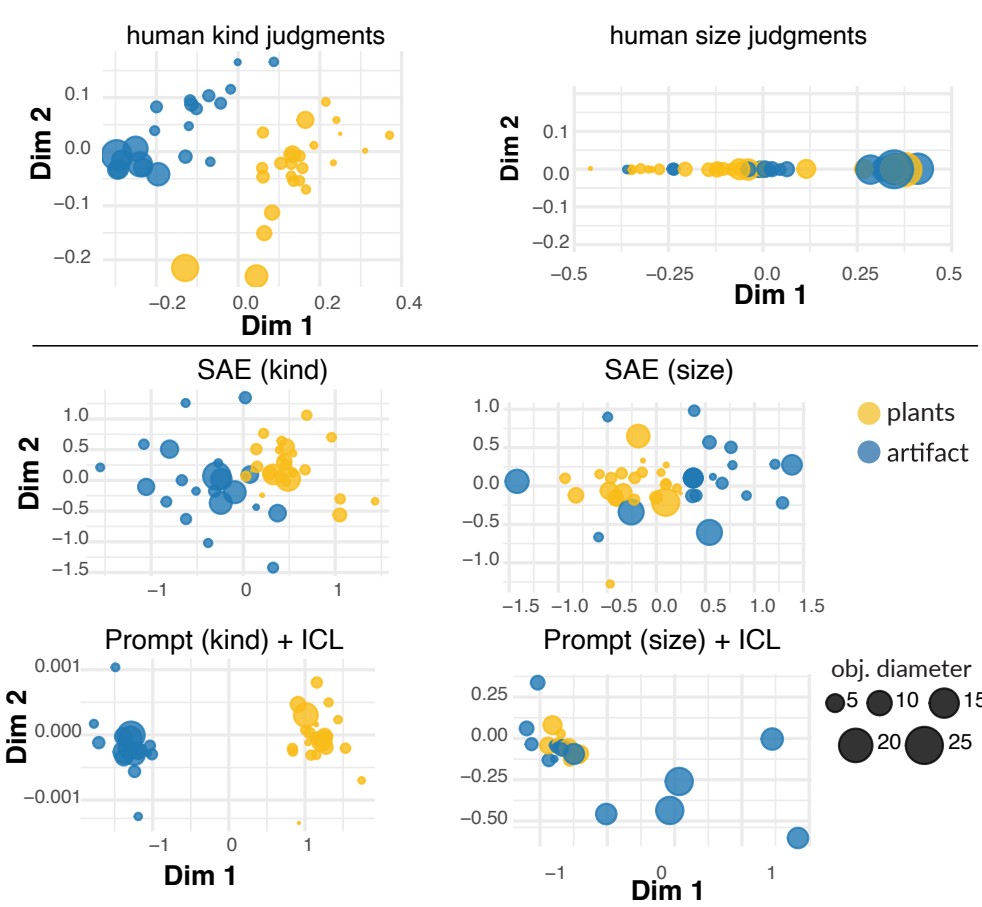

Figure 2: Representational geometry of the concepts in the Round Things Dataset based on embeddings derived from triadic judgments for humans (top row) and `gemma2-27b` (bottom 2 rows) using two steering methods. Plots for all methods can be seen in A.5

In the *kind* condition, prompting methods were also most effective in aligning model representations with humans. A regression predicting representational alignment from prompting revealed a significant improvement for prompting over non-prompting methods ($\beta = 0.23$, $t = 4.09$, $p = 0.003$), consistent with the large Procrustes correlations observed for all prompt-based methods ($r_{\text{prompt–kind–27b}} = 0.82$, $p < 0.001$; $r_{\text{prompt–kind–9b}} = 0.76$, $p < 0.001$; $r_{\text{prompt–ICL–27b}} = 0.76$, $p < 0.001$; $r_{\text{prompt–ICL–9b}} = 0.77$, $p < 0.001$). Steering methods showed weaker but in several cases still significant alignment ($r_{\text{TV–9b}} = 0.54$, $p = 0.015$; $r_{\text{DM–27b}} = 0.66$, $p = 0.001$; $r_{\text{DM–9b}} = 0.52$, $p = 0.022$; $r_{\text{SAE–9b}} = 0.59$, $p = 0.007$).

In the *size* condition, we observed no such benefit in the prompting conditions. The regression showed no prompting advantage ($\beta = -0.06$, $t = -0.65$, $p = 0.54$), consistent with the weak Procrustes correlations for zero-shot and in-context prompts ($r = 0.42$, $p = 0.078$ for 27B; $r = 0.28$, $p = 0.26$ for 9B; all prompt–ICL $r < 0.17$, $p > 0.49$). Interestingly, the only significantly aligned representations for *size* judgments were in the task vector condition in the smaller model ($r_{\text{TV–9b}} = 0.48$, $p = 0.037$).

## 5 CONCLUSION

We evaluated a set of popular LLM steering techniques using a well-established method in the cognitive sciences—triadic similarity judgments. By using both steered LLM and human judgments as input into embedding algorithms that capture an agent's representational geometry, we applied

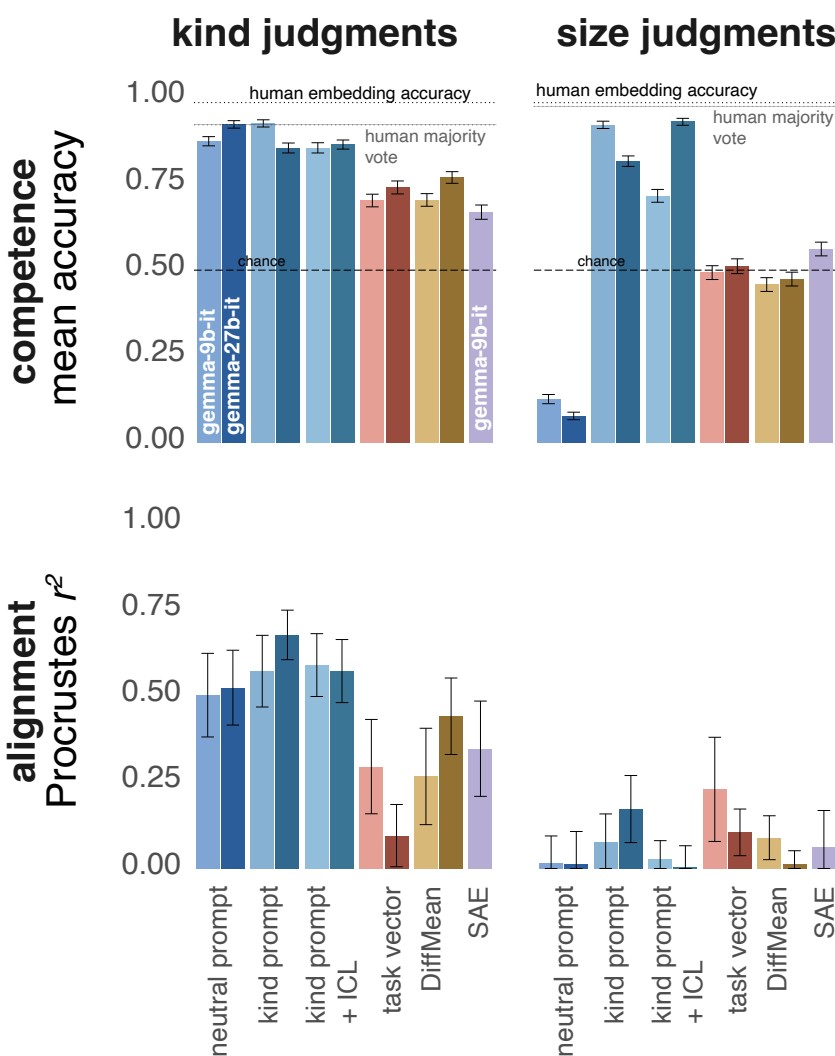

Figure 3: Steering accuracy (top row) and alignment of steered LLM representations to human representations (bottom row) for each steering technique. the dashed line labeled 'human embeddings' corresponds to how accurately human judgments can be predicted from human embeddings. SAE results are reported only for `gemma-9b-it` due to compute limitations.

a commensurate standard to evaluate how human-like different steering methods are. Critically, we looked beyond raw accuracy (competence) and evaluated steered models on how strongly their embeddings aligned with human embeddings. Similar to prior work (Wu et al., 2025), we found that prompting methods tended to outperform other steering techniques in terms of both accuracy and alignment. We also found that LLMs, without any steering, tended to be predisposed to privilege the *kind* dimension of similarity over the *size* dimension, indicated both by the stronger alignment of the neutral prompt condition with embeddings from human kind judgments and by the difficulty steered models had aligning with human size representations.

Taken together, these findings provide a novel perspective on how steering methods can and should be evaluated in order to assess how aligned steered model representations are to those of people. We find converging evidence that prompt-based steering is currently the best route for both accurate and aligned steering, and that some axes such as kind are privileged over size. Future work should seek to further integrate insights from controlled semantic cognition (Giallanza et al., 2024) to uncover the basis of prompting's success in guiding LLMs' learned representations in a context-sensitive manner, and to test a wider variety of contexts beyond size and kind judgments.

## 6 LIMITATIONS

**Model suite.** Due to compute constraints we only evaluated models that could be run on consumer-grade GPUs (`gemma2-9b` and `gemma2-27b`). Further, we did not evaluate `gemma2-27b` on the SAE method due to the lack of available trained SAEs for that model. Despite this, we believe the presented results constitute a fair comparison between methods. Future work can scale up this approach to larger models. The efficacy of different steering methods for both accuracy and alignment may scale differently with model parameter size. Between the 9B–27B scale we did not observe drastic changes, but this could change at larger scales.

**Method suite.** While we evaluated a representative set of steering methods for 'online' steering based on prompts or activation changes, we did not exhaustively test alternative 'in-weight' methods of changing model behavior (Anand et al., 2024), including supervised fine-tuning (SFT), low-rank adaptation (LoRA), and others. Since we are modeling human steering, which is thought to invoke online 'control' processes as opposed to in-weight learning (Giallanza et al., 2024), we argue that our chosen set of methods provided fair comparisons to the human data. Nevertheless, future work should compare the relative efficacy of in-weight vs. in-context styles of steering.

**Order effects.** While we randomized the order in which concepts were presented to the models for the triplet task—a common method in the behavioral sciences to protect against order effects—we note that we only ran the LLM evaluations with a single ordering. Since humans were also only presented with a single overall ordering, and our goal was to make commensurate comparisons across agent types, we chose to trade off robustness against order effects for fairer comparisons.

**Limited evaluation methods.** We relied on the triadic similarity judgment task since it is a well-established and validated method for evaluating human mental representations. In practice, there are more complex and naturalistic behaviors that humans perform that require 'steering' semantic representations (Giallanza et al., 2024). In the future, it will be crucial to incorporate such tasks when benchmarking different language model steering methods.

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

# A  STEERING METHOD AND EMBEDDING METHOD IMPLEMENTATIONS

## A.1  DETAILED IMPLEMENTATION OF STEERING METHODS

Let $f$ be a decoder-only language model with $L$ layers and hidden size $d$. Each triplet comparison is denoted as $t_i = (x_{\mathrm{ref},i}, x_{1,i}, x_{2,i})$ for $i = 1, \ldots, n$. Each prompt consists of a sequence of $n$ triplets $[t_1, \ldots, t_n]$, serialized into a token sequence $x = [x_1, \ldots, x_T]$. The model produces hidden states $h_j^l \in \mathbb{R}^d$ at each token position $x_j$ and layer $l$.

For all prompting, prompts are formatted as natural language strings of the form:

```
Choose the item that is most similar to the first item
in terms of <d>.  Respond only with the name of the
item exactly as written.
<x_ref> + <x_1> OR <x_ref> + <x_2>?
answer:  <x_ref> +
```

For steering methods, training examples consist of triplets without a natural language instruction like so:

```
<x_ref> + <x_1> OR <x_ref> + <x_2>?
<x_ref> + <x_answer>
```

Where examples are concatenated by commas. Each training example consisting of $n$ triplets has a final incomplete training instance like so:

```
<x_ref> + <x_1> OR <x_ref> + <x_2>?
<x_ref> +
```

This "+" token is used to extract and steer representations for a corresponding "+" token in the zero-shot test example, which is identical to the final training example:

```
<x_ref> + <x_1> OR <x_ref> + <x_2>?
<x_ref> +
```

Fields are interpolated for some $d \in \{size, kind, neutral\}$, triplet $t_n = (x_{\mathrm{ref}}, x_1, x_2)$, and answer $t_{answer}$ given the dimension $d$. We extract activations, logits, and apply all steering methods at the final input token $x_T = +$ in the last triplet $t_n$, depending on each method.

### ZERO-SHOT PROMPT

In the zero-shot condition, the model is given a single triplet $t_n = (x_{\mathrm{ref}}, x_1, x_2)$ and is asked to make a discrimination along a semantic dimension $d \in \{size, kind, neutral\}$.

### PROMPT WITH IN-CONTEXT EXAMPLES

In the in-context condition, the model is given a sequence of $n = 15$ complete triplets $[t_1, \ldots, t_{15}]$ and is asked to make a discrimination for the final triplet $t_{15} = (x_{\mathrm{ref}}, x_1, x_2)$ along semantic dimension $d \in \{size, kind\}$.

### TASK VECTOR

Following Hendel et al. (2023), we extract *task vectors* for the KIND and SIZE conditions by first constructing two prompts organized along each condition: $x_{\mathrm{train}}$, containing 14 complete triplet examples and one final incomplete example (with `"+"`), and $x_{\mathrm{test}}$, containing a single zero-shot incomplete triplet.

For each layer $\ell \in \{0, \ldots, L\}$, we extract the hidden activation in the residual stream at the final token position of $x_{\mathrm{train}}$ (i.e., the `"+"`) and patch it into the corresponding position in $x_{\mathrm{test}}$. The language model $f$ then autoregressively generates a sequence $x_0, \ldots, x_k$ until a complete output is produced.

We repeat this procedure over 200 randomly generated $(x_{\text{train}}, x_{\text{test}})$ pairs, selecting the layer $\ell_d^*$ that yields the highest accuracy. Finally, using this optimal layer $\ell_d^*$, we repeat the procedure across 2400 additional prompt pairs to generate task vector embeddings for both the SIZE and KIND conditions.

DIFFMEAN

DIFFMEAN constructs a steering vector by computing the average difference between latent representations of *positive* and *negative* examples along a target output dimension. Specifically, the mean latent representation of the positive examples is subtracted from that of the negative examples, producing a steering vector. This vector is then *added* (as opposed to task vectors, where it is patched) to the latent representation of a held-out prompt $x_{\text{test}}$ in order to steer the model's output generation.

For a target steering dimension $d \in \{size, kind\}$ and its contrast $d'$, we generate 15 triplet examples organized along $d$, and 15 along $d'$, where the final triplet in each set is incomplete and ends with the `"+"` token.

Then, for a given layer $\ell \in \{0, \ldots, L\}$, we extract the residual stream representation $r_T^\ell$ at the final token position $T$ from both $x_{\text{train},d}$ and $x_{\text{train},d'}$. The DIFFMEAN steering vector is then computed as the difference:

$$v_{\text{diff}} = r_T^\ell(x_{\text{train},d}) - r_T^\ell(x_{\text{train},d'})$$

This resulting vector $v_{\text{diff}}$ defines a direction in the residual stream corresponding to the contrast between the dimensions $d \in \{size, kind\}$.

Similar to the task vector condition, We repeat this procedure over 200 randomly generated $(x_{\text{train}}, x_{\text{test}})$ pairs, selecting the layer $\ell_d^*$ that yields the highest accuracy. Finally, using this optimal layer $\ell_d^*$, we repeat the procedure across 2400 additional prompt pairs to generate DiffMean embeddings for both the SIZE and KIND conditions.

SAEs

We use sparse autoencoders (SAEs) from GEMMASCOPE (Lieberum et al., 2024) to steer the model along interpretable directions in residual space. An SAE is a linear model that decomposes a residual stream vector $r \in \mathbb{R}^d$ as a sparse linear combination of features, where $W \in \mathbb{R}^{d \times k}$ is a learned feature dictionary and $z \in \mathbb{R}^k$ is a sparse activation vector. Each column of $W$ defines a directional feature in residual space, and only a small number of features are active for any given input.

For each semantic dimension $d \in \{size, kind\}$, we construct 20 prompts and identify the feature $f \in \mathbb{R}^d$ with the highest average activation at layer $\ell = 20$. We steer the model by injecting $c \cdot f$ (with $c = 50$) into the residual stream at layer 20 (the only available layer for `gemma-2-9b-it` on GEMMASCOPE), and generate 2400 zero-shot completions from held-out prompts $x_{\text{test}}$. These completions are used to construct semantic embeddings for each SAE-steered dimension.

## A.2 PROCRUSTES CORRELATIONS FOR ALL PROMPT AND STEERING METHODS

The comprehensive set of procrustes correlations for all steering methods, for `gemma-2-9b-it` and `gemma-2-27b-it`.

Figure 4: Full procrustes correlations for all methods

## A.3 ICL PROMPT ANALYSIS

We systematically vary the number of example triplet pairs included in the KIND condition for `gemma-2-9b-it` to examine how changes to the input prompt affect model accuracy and human alignment. We find that there is no significant impact of the number of ICL examples on accuracy or alignment.

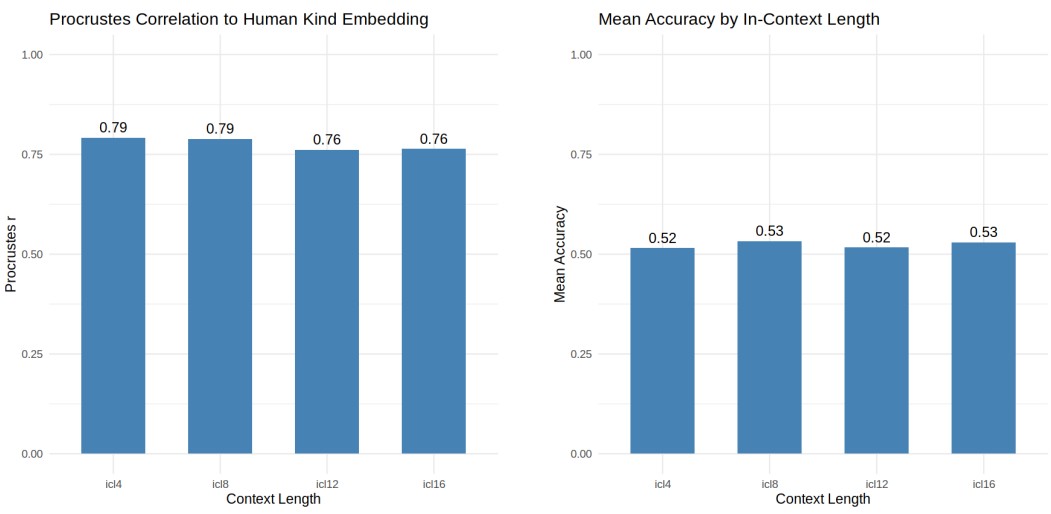

(a) Procrustes correlations with varied ICL examples    (b) Mean accuracy with varied ICL examples

Figure 5: Impact of varying the number of ICL examples on model performance

## A.4 EMBEDDINGS DIMENSIONS ANALYSIS

Cumulative variance explained by the first $k$ dimensions of the embeddings for `gemma-2-27b-it`, size condition. The first two dimensions of all embeddings were used for comparisons of representations.

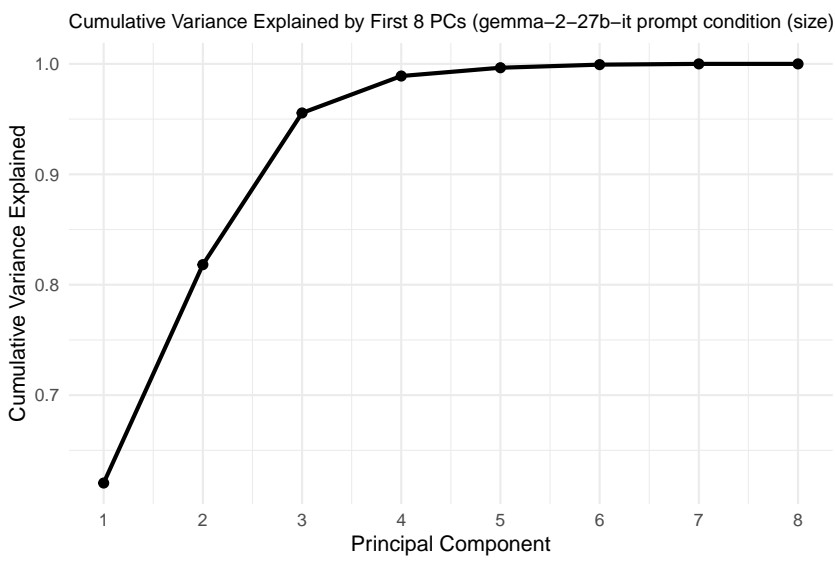

Figure 6: Cumulative variance explained by embedding dimensions

## A.5 FULL EMBEDDING PLOTS

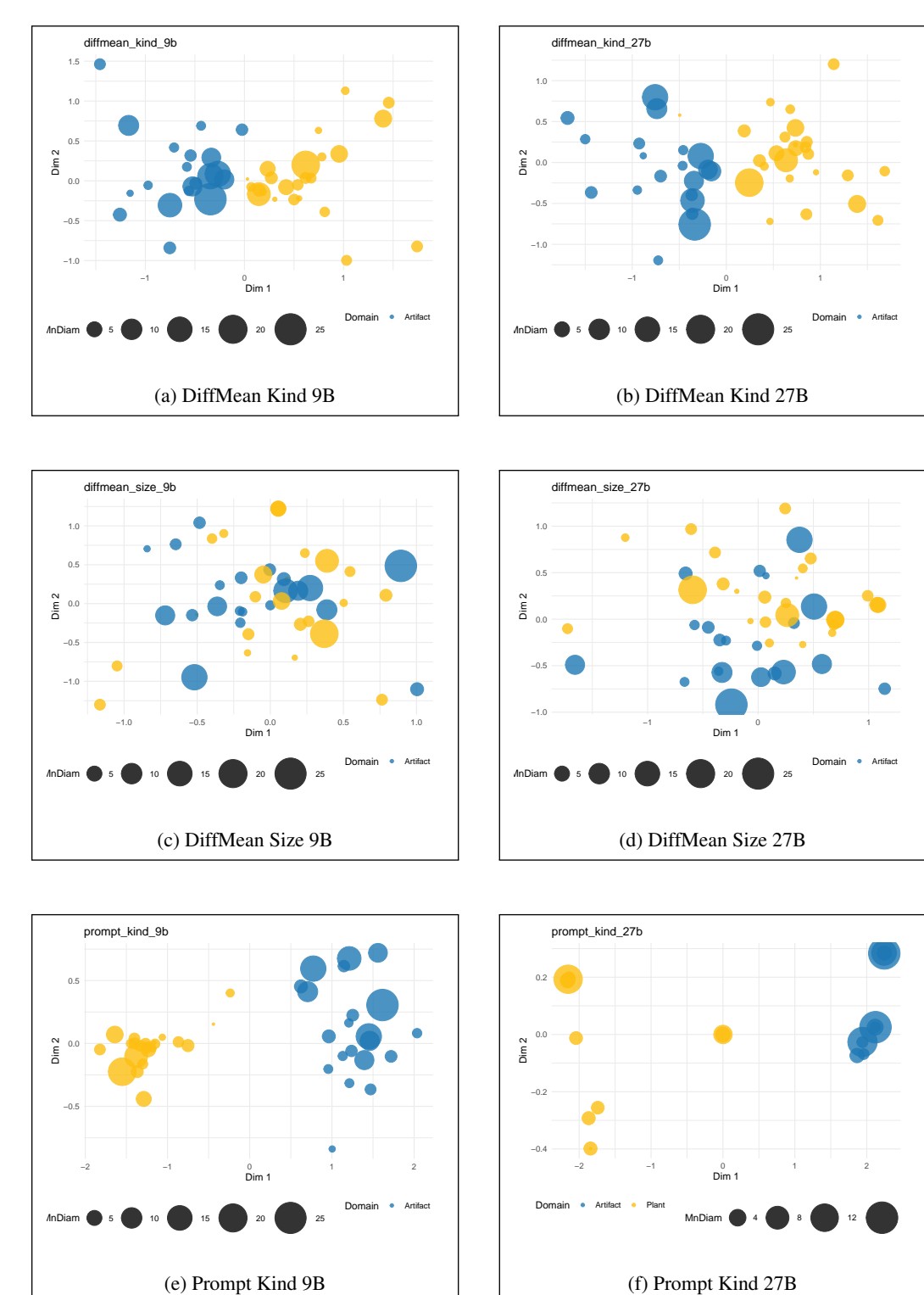

Figure 7: Embedding Plots: DiffMean and Prompt Methods

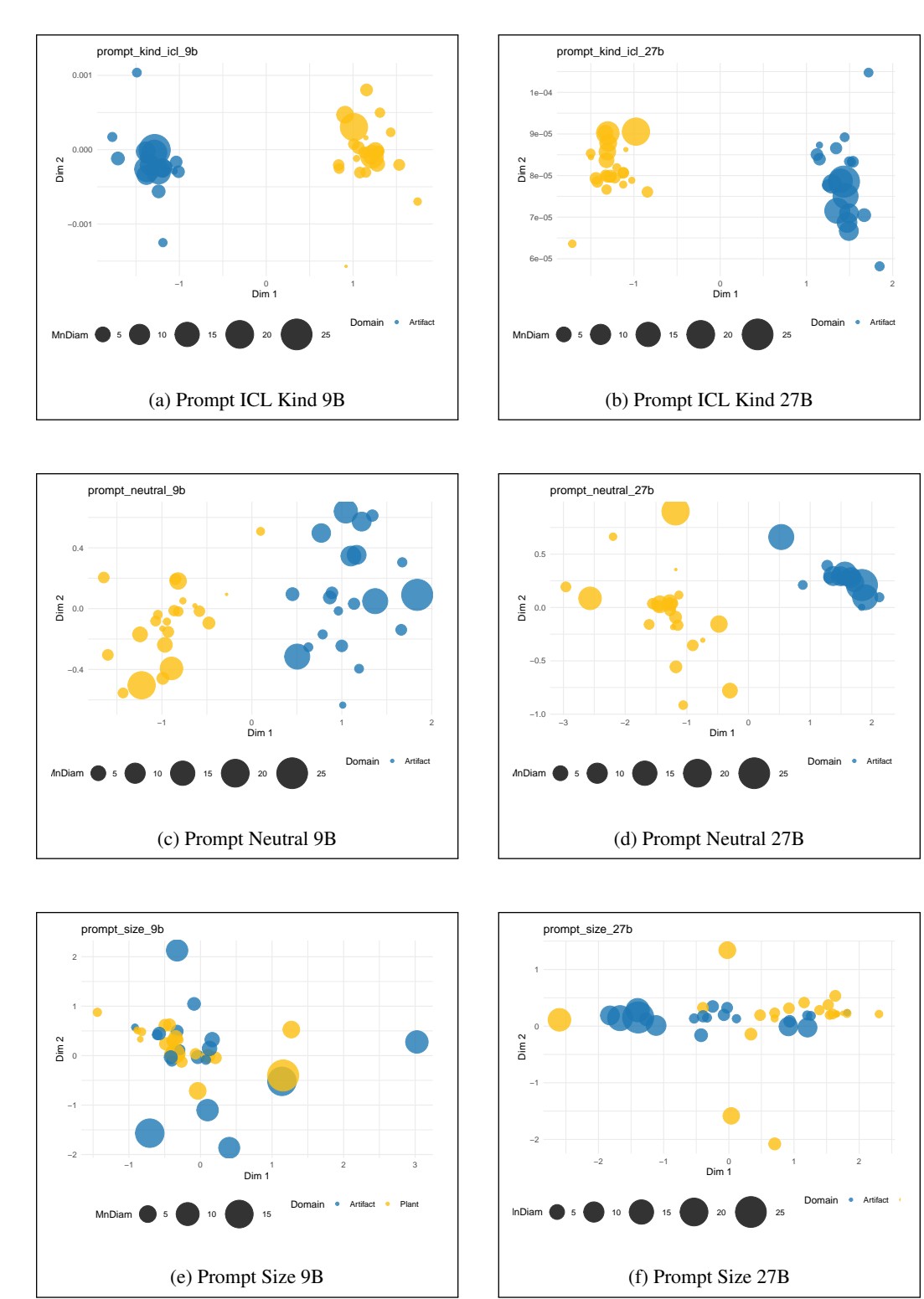

Figure 8: Embedding Plots: Prompt Method Variations

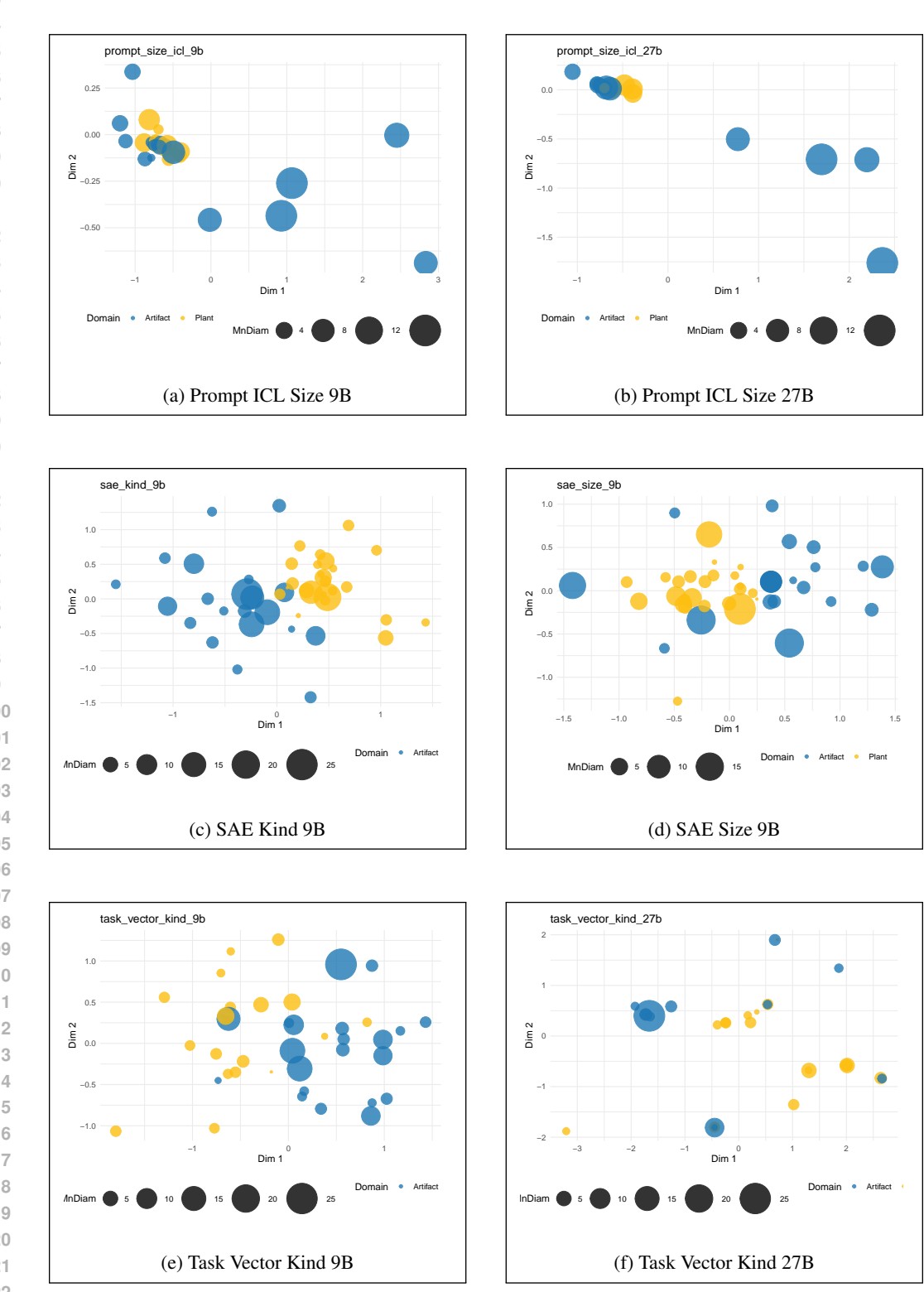

Figure 9: Embedding Plots: Prompt ICL, SAE, and Task Vector Methods

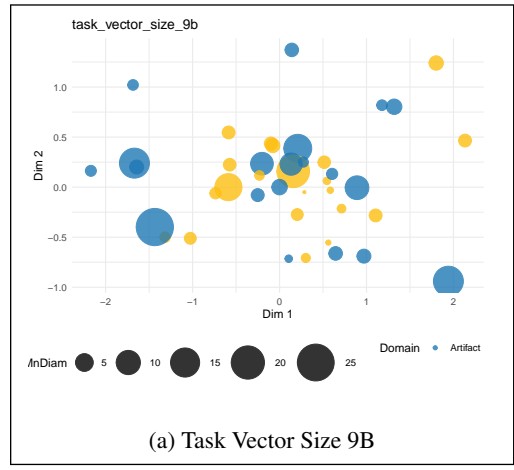 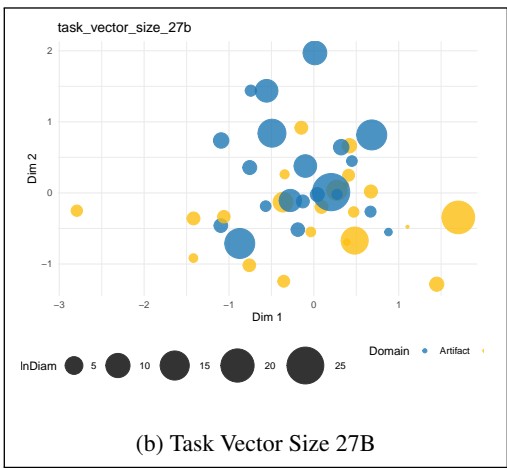

(a) Task Vector Size 9B     (b) Task Vector Size 27B

Figure 10: Embedding Plots: Task Vector Size Condition

