# OpenReview forum: "Evaluating steering techniques using human similarity judgments"
_ICLR.cc/2026/Conference — ICLR 2026 Conference Desk Rejected Submission_

### Official Review · Reviewer_mVQ2 · 2025-10-29

**Soundness:** 2
**Presentation:** 3
**Contribution:** 2
**Rating:** 6
**Confidence:** 3

**Summary:**

The paper proposes a cognitively grounded evaluation of LLM steering methods using triadic similarity judgments. The task asks models to choose which of two items is more similar to a reference along an instructed dimension (size or kind). From many triplet judgments, the authors reconstruct two-dimensional embeddings using a crowd-kernel approach and then quantify model–human representational alignment via Procrustes correlation. Across Gemma2-9B and Gemma2-27B, prompt-based steering achieves the highest task accuracy and the best, though still limited, alignment; models show a default bias toward kind over size; and none of the interventions produce embeddings that align well with human size representations. The work argues for evaluation beyond task accuracy, emphasizing alignment between steered model representations and human cognition.

**Strengths:**

The experimental setup is coherent:
- define triplets with mutually exclusive size vs kind decisions
- collect at least 2,500 judgments per method
- fit 2D embeddings with the crowd-kernel loss, and
- compute squared Procrustes correlations between human-derived and model-derived embeddings.

The instruction formats, in-context learning vs zero-shot, and activation-based interventions are described clearly in the appendix with a consistent steering and evaluation pipeline. The separation of competence (accuracy on held-out triplets) and alignment (geometry similarity to human embeddings) is appropriate and addresses the common “performance vs representation” conflation.

The competence–alignment dissociation is carefully documented: prompting yields high task accuracy, yet alignment, especially for size, remains low. The kind-over-size default is convincingly supported by higher neutral-prompt accuracy and higher neutral-kind alignment relative to size. The paper’s hypothesis that humans allow leakage from irrelevant dimensions while well-prompted LLMs can more cleanly isolate task-relevant features is plausible and consistent with observed dissociations, although corroborating analyses would strengthen it. For example, measuring off-axis interference directly by training a linear probe on model residual states to predict kind from size prompts (and vice versa) would quantify leakage, and doing the same on human embeddings would create an explicit comparison.

The narrative fits with emerging evidence that simple prompting often rivals or beats more complex internal interventions and that model competence need not imply human-like internal representations; the paper’s added value is to anchor this in an interpretable task with a geometry-level alignment metric.

**Weaknesses:**

- The decision to fix the embedding dimensionality at two could artificially compress structure and differentially affect methods; it would be helpful to report results across multiple dimensionalities with model selection via held-out triplets or information criteria, or to show that conclusions are stable at d=3–5.
- Procrustes r^2 should be accompanied by uncertainty estimates, for example via bootstrapping triplets, and significance assessed against a permutation baseline that preserves triplet structure to rule out spurious correlations.
- The selection of layers for the activation-based methods appears to be tuned on held-out accuracy and then reused for alignment estimation; a fully nested cross-validation would avoid any potential selection–evaluation coupling.
- The parsing of model outputs requires precision to avoid leakage from surface-form priors; please detail how exact string matching was enforced and whether alternative surface forms (e.g., synonyms, capitalization) were encountered and handled.

**Questions:**

- How many human participants contributed to each condition, what were per-participant trial counts, and how were judgments aggregated? Were within-participant reliabilities assessed, and is there an upper-bound “human-to-human” Procrustes r^2 to contextualize model alignment?
- How sensitive are results to embedding dimensionality and to the choice of optimizer and regularization in the crowd-kernel fitting?
- For activation-based methods, were multiple layers or compositions considered at inference time beyond the best single layer, and does multi-layer steering change the story?

**Details Of Ethics Concerns:**

The study raises minimal direct ethical risks. It uses human judgments; the paper should explicitly state the consent procedure, compensation, and IRB/ethics approval if applicable. Since the work evaluates steering, it might indirectly influence how practitioners attempt to control models; discussing dual-use considerations and emphasizing that alignment to human-like representations is not synonymous with normative safety would be valuable.

---

> ### Author Response · Authors · 2025-11-26
> **Response to reviewer mvQ2**
>
> We thank the reviewer for their thoughtful response to the paper and for raising several valid concerns which we have addressed in the results and discussion below. We first respond to each of the weaknesses raised and then the questions posed by the reviewer.
>
> Regarding weakness 1 (the decision to fix dimensionality at d=2). We agree that using two dimensions for model embeddings could potentially compress model representations. Although we are limited to two dimensions in human data (where participant inference is costly) it is indeed important to ensure that our model representation findings are not an artifact of low dimensionality. To address this, we fit embeddings for dimensionalities d=2…6, and evaluate test loss and accuracy for each embedding. This allows us to see whether lower dimensionality fails to capture the full representational structure of model judgments. We find that d=2 already performs well on a held out test set of triplets (accuracy=0.956), and that there is no improvement to test accuracy when increasing dimensionality. We observe a slight decrease in test loss from d=2 to d=6, although the absolute difference in performance is less than 0.01. Taken together, we believe these results support using two dimensions in analyzing model representations, especially as this is commensurate with existing human data. An image of the plot for test loss for dimensions 2…6 is attached anonymously below.
>
> https://imgur.com/a/ohY56pB
>
>
> Regarding weakness 2 (uncertainty estimates for Procrustes r^2): this is an excellent idea, and we agree that a more rigorous analysis of procrustes correlations will help to rule out spurious results. We estimated standard errors for each Procrustes r^2 relation using a permutation test, and added these as error bars to figure 3. Specifically, we estimate a null distribution by permuting labels(human vs. model-condition-x) for all relations, and then compute the standard error of the observed R2 relative to this null distribution. An updated version of the Procrustes R2 alignment portion of figure 3 can be previewed here, and will be added in the updated version of the submission.
>
> https://imgur.com/gallery/procrustes-r2-alignment-plot-kind-size-with-error-bars-OBSOhY6
>
>
> Regarding weakness 3 (cross validation for layer selection) our goal was to evaluate common steering methods as they are generally implemented in the field. As a result, we defer to Hendel et al. (2023) and Ilharco et al. (2023) for best practices in the implementation of task vectors, and Marks & Tegmark (2024) for DiffMean. By systematically sweeping across layers, we “overfit” on task vector and difference-of-means performance, but also give each method its best chance of succeeding on our task. In spite of this, these methods still show low human-alignment in their semantic representations on this extremely simple task, which we take as evidence of existing disparities in the extent to which steering methods can align to human representations.
>
> Regarding weakness 4, we take several steps to ensure that model responses which contain the right answer are counted as correct. Specifically, we extract the first token of the target response (e.g “beach” in “beach ball”) and do a case-insensitive match for this partial string anywhere in the model string output. We also generate seven token outputs from each model to ensure that minor errors in instruction-following are permissible (such as the model beginning a response with “the answer is probably…”). If the string partial is in the seven token sequence, the answer is counted as correct. Overall, the objective is to solicit semantic representations from all models and steering representations in a way that tests semantic knowledge itself, as opposed to issues with case sensitivity or instruction following.

---

> ### Author Response · Authors · 2025-11-26
> **Response to questions from reviewer mvQ2**
>
> Q1. Excellent question! We provide the summary statistics for the human data below, which are also detailed in Giallanza et al. (2024). 39 participants completed the "kind" condition (contributing 3,887 judgments) and 40 participants completed the "size" condition (contributing 3,3944 judgments). Participants performed the task for either 5 minutes or 100 trials, whichever came first. The mean number of trials per participant was 98.67 (SD = 1.66) in the kind condition and 98.60 (SD = 4.53) in the size condition.
> Judgments were aggregated using the Crowd Kernel ordinal embedding algorithm (Tamuz et al., 2011).The embedding was trained on 90% of the data selected at random, with the remaining 10% used as a test set. The final two-dimensional embedding was selected based on the lowest observed test-set loss across 30,000 training epochs.
>
>
> We note that due to the per-participant trial counts, it is not possible to fit individual human embeddings given the theoretical lower bound on observations for the crowd-kernel algorithm, as stated in Sievert et al. (2023). Fitting individual human embeddings is an interesting empirical question which we leave open to future work.
>
> Q2. This is a great question, and we agree that it is important to ensure that our results are not an artifact of sub-optimal embeddings dimensionality. We refer the reviewer to our response to weakness 1, where we systematically evaluate loss and held out test accuracy on triplets for varying dimensionalities (d=2…6). The plot for test loss accuracy across dimensions can be seen again here.
>
> https://imgur.com/a/ohY56pB
>
> Q3. Multi-layer steering is indeed an interesting future direction! In our case, the extensive hyperparameter search to find the right layer combinations is computationally intractable, given how the existing computational demands of systematically sweeping for the optimal single-layer intervention would scale up to all possible layer-layer combinations.

---

### Official Review · Reviewer_vtyN · 2025-10-31

**Soundness:** 3
**Presentation:** 3
**Contribution:** 3
**Rating:** 8
**Confidence:** 3

**Summary:**

This paper evaluates the efficacy of Large Language Model (LLM) steering techniques by assessing how well the steered representations align with human derived representations. Both humans and LLMs were asked to judge which of two concepts was most similar to a target concept, along kind and size dimensions. LLMs were steered using 4 techniques and measured on accuracy and alignment. Prompting outperformed other steering methods. The work also discovered that LLMs are biased towards similarity judgements based on kind, and struggle with size. Outcome is that prompting is the most effective steering approach and LLMs exhibit predisposed representational axes i.e. kind vs. size.

**Strengths:**

The study moves beyond task specific accuracy to evaluate the representational alignment of steered LLMs with human cognition.

Application of triadic similarity judgment task to both humans and LLMs and comparing results.

The use of the Round Things Dataset allows for a controlled investigation of how task context selectively emphasizes one dimension over the other.

Comprehensive comparison of steering methods.

Discovery of inherent LLM bias towards a specific axis over another via the inclusion of neutral prompt.

**Weaknesses:**

The study relies exclusively on triadic similarity judgment, this is good for controlled isolation, but the results may not generalize to more complex application.

Would be interesting to see how larger models evaluate.

Order effects. The authors mention that the experiments were run using only a single overall ordering. I understand the reasoning but given the knowledge that LLMs do suffer from ordering bias, it would have been better to at least report on results that included a mitigation for this phenomenon.

**Questions:**

Any expectations on how the experiments would perform on larger models, or models with different architectures?
Any theory as to why alignment with human size judgments was consistently poor?
Comments/theory as to why prompting was the most successful technique?

---

> ### Author Response · Authors · 2025-11-26
> **Response to reviewer vtyN**
>
> We thank the reviewer for their comments and for raising opportunities for improving our paper! We first respond to each of the weaknesses raised and then the questions posed by the reviewer.
>
> Regarding weakness 1 (limitations of the triadic similarity judgment task). We agree that on face value this appears to be a relatively simple task, but we wish to emphasize that the ability to produce accurate/human-like judgments on this task has been thought to tap into deep semantic knowledge (Hebart et al.,2023; Muttenhaler et al., 2025; Giallanza et al., 2024). There is a deep literature on the use of this kind of task in assessing quite generalized semantic knowledge in humans and machines. In fact, recently it has been shown that optimizing models using the triadic similarity judgment task leads vision models to learn more human-like visual representations, which we feel is strong evidence for the generalizability of this approach (Muttenhaler et al., 2025; ).
>
> References:
>
> Muttenthaler, L., Greff, K., Born, F., Spitzer, B., Kornblith, S., Mozer, M. C., ... & Lampinen, A. K. (2025). Aligning machine and human visual representations across abstraction levels. Nature, 647(8089), 349-355.
>
> Giallanza, T., Campbell, D., Cohen, J. D., & Rogers, T. T. (2024). An integrated model of semantics and control. Psychological Review.
>
> Hebart, M. N., Contier, O., Teichmann, L., Rockter, A. H., Zheng, C. Y., Kidder, A., ... & Baker, C. I. (2023). THINGS-data, a multimodal collection of large-scale datasets for investigating object representations in human brain and behavior. Elife, 12, e82580.
>
>
> Regarding weakness 2 (evaluating larger models), we agree that it is helpful to know how generalizable findings are as model scale increases. This was what motivated us to test a smaller and larger gemma model. As noted in response to reviewer FSmf, it is computationally expensive to run the steering evaluations at scale, even for two models given our resources, and we would not be able to efficiently train SAEs for models larger than 27b. However, as a compromise, we do try to establish baselines for non-activation based steering methods using the frontier model OpenAI o3 (ostensibly much larger). We evaluate o3 on the prompt condition for both size and kind judgments, and observe interesting convergences with our findings in smaller models. Notably, while o3 accuracy is high for both kind and size judgments (0.99 and 0.93, respectively) we see a large disparity in human alignment between the kind and size conditions: while o3 kind representations are relatively well aligned with those of humans ($R^2$ = 0.81), size representations show a poor degree of alignment ($R^2$ = 0.24). We believe these results demonstrate the generalizability of this task to larger models: even at scale, frontier models remain misaligned with humans along even basic semantic dimensions.
>
> Plots for these results can be seen here, and will be included in the final appendix.
>
> https://imgur.com/gallery/o3-frontier-model-human-alignment-AGjklsn
>
>
> Regarding weakness 3 (order effects). We agree that it is worthwhile to test for sensitivity for ordering effects, we have now included an analysis where we systematically flip the order in which options were presented and compare the accuracy/alignment to what we find in the original results. We run gemma-2-9b-it once with results in the original order, and once with all options (a,b) inverted. We observe high consistency between runs, with model choices remaining consistent in 0.886 of all trials. Moreover, overall model accuracy remains consistent across both runs ( 0.926 ground truth-accuracy for original ordering, 0.921 for the inverted ordering). We also evaluate the inverted model judgments in terms of human alignment, and find that the procrustes $R^2$ with human judgments to be 0.121 for the inverted trials, compared to 0.076 for non-inverted trials. While there is evidently some degree of noise depending on trial order, the overall similarities in responses, accuracy, and alignment between inverted and non-inverted trials suggests general robustness in our overall results.

---

> ### Author Response · Authors · 2025-12-03
> **Response to questions posed by reviewer vtyN**
>
> Any theory as to why alignment with human size judgments was consistently poor?
>
> We suspect this stems from fundamental differences in how humans and transformers represent semantic knowledge. In humans, there’s evidence for a “semantic hub” - a modality-invariant representational system that stores general conceptual knowledge, which then gets flexibly transformed based on task demands through cognitive control mechanisms (Patterson et al., 2007; Rogers et al., 2004). This architecture means that even when humans perform a size judgment task, their representations still reflect broader semantic structure (e.g., taxonomic relationships, other feature dimensions) - they’re accessing general conceptual knowledge and selectively attending to task-relevant dimensions.
> Transformers, by contrast, appear to lack this separation between stable semantic representations and task-driven modulation. The attention mechanism seems to “hypercontextualize” - aggressively reshaping representations to optimize for the immediate task rather than drawing on a general semantic store. This would explain why these models achieve high behavioral accuracy on size judgments while showing poor representational alignment with humans: they’re solving the task through a fundamentally different computational strategy that doesn’t preserve the rich, multidimensional semantic structure humans maintain even during focused judgments.

---

> ### Author Response · Authors · 2025-12-03
> **Response to questions posed by reviewer vtyN (part 2)**
>
> Comments/theory as to why prompting was the most successful technique?
>
> The success of prompting likely reflects the extensive post-training these models undergo to follow instructions. Modern LLMs are heavily optimized through instruction tuning, supervised fine-tuning, and RLHF to interpret and comply with natural language directives. This creates a well-developed “interface” for steering model behavior through prompts, whereas techniques like activation steering or representation engineering operate on internal mechanisms that weren’t explicitly optimized for external modulation. In essence, prompting works with the grain of how these models were trained to be controlled, rather than against it.

---

### Official Review · Reviewer_FSmf · 2025-11-01

**Soundness:** 3
**Presentation:** 3
**Contribution:** 3
**Rating:** 6
**Confidence:** 4

**Summary:**

The paper evaluates LLM steering methods (prompt and activation) on Gemma-2 models using a similarity task on the Round Things dataset. Prompting gives the highest accuract and strongest alignment to human reps. Unsteered models default to kind-based similarity and are not well-aligned.

**Strengths:**

+ I like the question of evaluating whether steering interventions produce human-like representations

+ The task design seems carefully constructed, and the implementation details are quite extensive

**Weaknesses:**

- Accuracy appears to be defined against decisions induced by the fitted human embedding rather than some other more reasonable measure like majority human response on the same triplet. Is this circular?

- The experiments are lacking some important details about statistics, specifically the results in Section 4 don't appear to specify the underlying model (is it logistic regression?), confidence intervals, how multiple comparisons are handled, etc.

- Experiment is focused only on two Gemma-2 models. Also, steering does not include simple interventions like LoRA or SFT which would help to separate if prompting's advantage is just instruction-following ability rather than a limitation of activation steering

**Questions:**

- Can the authors report model accuracy against majority human triplet responses in addition to the current embedding labels?

- Were the 200 prompt pairs used for layer selection in task vectors and DiffMean disjoint from the 2400 pairs used for final evaluation?

- How sensitive are the results to the chosen injection layer and magnitude?

---

> ### Author Response · Authors · 2025-11-26
> **Response to reviewer FSmf**
>
> We thank the reviewer for their close reading of our paper and for highlighting several actionable changes that we believe strengthens the core paper. We first respond to each of the weaknesses raised and then the questions posed by the reviewer.
>
> Regarding weakness 1 (accuracy being defined w.r.t. human embeddings). First, we would like to clarify that accuracy (Fig 3. First row of graphs) is computed with respect to ground truth size and kind values that are documented in the Round Things dataset. It is precisely this neat ground truth that motivated the use of this dataset to evaluate the efficacy of steering. The reviewer is correct that we include a dashed line for ‘human embeddings’ in these plots that shows how well human embeddings predict the ground truth. We include this because even humans might not be perfect at ascertaining the ground truth (which they are not, since the dashed line is not at 1.00), and we want to contextualize model performance with respect to this baseline. That being said, the reviewer’s idea to use the human majority vote is also an excellent idea since sometimes populations of human decisions can be more reliable than individual judgements or perhaps a single set of embeddings (e.g., the wisdom of crowds effect). We have now included a new dashed line in Figure 3 (top row) reporting what the human majority vote accuracy is as well. We hope this addresses the weakness noted by the reviewer. We will update the main manuscript with this figure, but for now we include this (anonymized) image of the graph for the reviewers’ convenience –
>
> This updated version of figure 3 can be seen here:
>
> https://imgur.com/gallery/figure-3-accuracy-plot-updated-with-majority-vote-accuracy-baseline-C3OHjJa
>
> Regarding weakness 2 (lack of statistical analysis details), we have added several new details in section 4  describing statistics. We briefly include some of the major changes below here, but are open to addressing any further questions the reviewer has on this front.
>
> We will update the main pdf in the coming days, but here is the update text for the reviewers convenience for 3 key instances where we also found our initial submission to be lacking in details -
>
> (1) We established two key baselines – how reliably embeddings estimated on human responses could predict held out triplet trials and how much agreement there was among participants when shown the same triplets (a majority vote metric).
> In terms of accuracy, we found that no method reached the level of majority vote agreement or embedding estimated responses. (0.854 for prompt-kind 27b-it, 0.925 for prompt-kind, 9b-it; 0.816 for prompt-size 27b-it, 0.921 for prompt-size, 9b-it;), however prompt-based methods approached the human majority vote baseline, reaching this threshold in the case of the `kind’ task (0.925, prompt kind, 9b-it).
>
> (2) To evaluate whether models were predisposed to be better at size vs. kind judgments when prompted, we fit a logistic regression model predicting models’ accuracy (based on size or kind ground truth values) from a categorical predicting the code for whether a trial was size or kind trial. We observed a higher accuracy of neutral prompts for kind judgments relative to their lower accuracy for size judgments (β = 0.187, p < 0.001), suggesting that models struggle when it comes to size judgments and are predisposed to answer based on semantic category. We found a similar trend for alignment, where we first computed the overall Procrustes correlation between models and human embeddings and then computed a Fisher r-to-z transform to test whether these alignment values varied between the size and kind conditions. We did indeed find statistically significantly stronger alignment in the kind condition relative to the size condition (r neutral, kind = 0.25 vs. r neutral, size = 0.14, p < 0.001)
>
> (3) We wanted to further test for differences among intervention methods in terms of steering models towards the ground truth values. We once again used regression models to model the accuracy based on a categorical prediction coding for the different intervention methods, with prompting as the baseline method. We found that non-prompt based methods (SAEs, task vectors, and DiffMean) consistently performed worse than prompting (βTV = −0.29, βDM = −0.30, βSAE = −0.29, p < 0.001), particularly for kind judgment, which was indicated by a main effect of a predictor coding for the triplet trial type (size or kind) (βkind = 0.19, p < 0.001).
>
> We will continue to improve the prose of the results until the end of the discussion period, but we hope these clarifications in addition to more results about the triplet experiment methods. See responses to reviewer mvQ2 for more information on the methods for collecting human data.

---

> ### Author Response · Authors · 2025-11-26
> **Continued response to reviewer FSmf**
>
> Regarding weakness 3 (narrow set of models). While we do agree that we do not have a large model suite, we emphasize the focus here was on breadth of techniques versus breadth of models. Further we were conscious to include two models that varied reasonably in model size to ensure our results were not overindexed to a particular model scale. Given that it is computationally expensive to implement all the steering methods at scale even for a single model (especially training sparse autoencoders), we hope the reviewer understands why we did not evaluate a larger set of models. In response to both reviewer FSmf and vtyN's comments about model suite, we now include results for the prompt-based conditions on OpenAI o3, a frontier reasoning model that is likely of much larger scale than the gemma models and we replicate the core finding that models are by default 'kind' sensitive relative to size.
>
> While we agree that exploring fine-tuning methods like SFT and LoRA are valuable, we note that the focus of the current paper is on methods that allow for steering of pretrained models with no changes to model parameters – drawing on critical analogies to cognitive control in human minds. We will now further explicitly emphasize the limitations of the paper only exploring ‘online’ evaluation methods and explain why we do so.
> Specifically for the limitations section Method Suite. we will add -
> It may be valuable in future work to evaluate the relative contributions of in-weight (finetuning) vs. in-context (online) learning and identify the cases where one method outperforms the other, as naturalistic learning in humans might involve a mix of cognitive control with updates to learned representations.

---

> ### Author Response · Authors · 2025-11-26
> **Response to questions from reviewer FSmf**
>
> We now turn to the questions raised by the reviewer.
>
> Q1. Yes, as noted above, we believe that addition of human majority vote baselines is a valuable contribution and we have now included dashed lines corresponding to this metric to Figure 3. and corresponding text discussing the accuracy w.r.t. this baseline.
>
> We established two key baselines – how reliably embeddings estimated on human responses could predict held out triplet trials and how much agreement there was among participants when shown the same triplets (a majority vote metric).
> In terms of accuracy, we found that no method reached the level of majority vote agreement  (0.854 for prompt-kind 27b-it, 0.925 for prompt-kind, 9b-it; 0.816 for prompt-size 27b-it, 0.921 for prompt-size, 9b-it;) However, prompt-based methods approached the human embedding baseline, reaching this threshold in the case of the `kind’ task. (0.925, prompt kind, 9b-it).
>
>
> Q2. While these prompts were not disjointed, these were randomly seeded so as to not overindex on specific pairs. Our goal was to implement these methods as they’ve been established and this was the approach taken by Hendel et al. (2023). Since our goal here is to find the ‘best’ layer we care less about strict cross-validation and one might imagine that in the limit we find the best layer for each prompt on a case-by-case basis, which would be computationally intractable. We argue that by selecting the layer and coefficient hyperparameters to give each steering method the best possible performance, we offer a “best case scenario” estimate of each method. The finding that these steering methods still significantly underperform in human-alignment despite systematically selecting the optimal layer for each method highlights an interesting disparity in the effectiveness of these methods on even simple tasks.
>
> Q3. We think this is an excellent question and once again, we followed the best practices established by the field here, but we agree that it is scientifically interesting to understand what the choice of layer and steering coefficient has on accuracy/alignment. We have described our approach for choosing the best layer (we would probably not want to choose suboptimal layers), and in agreement with the concerns of the reviewer, we also systematically test the difference of means steering condition for a range of steering coefficient values, to understand whether the coefficient value itself might be suboptimal. We evaluate steering coefficients of 1.0, 1.2, 1.4, and 5.0, and observe almost no difference in accuracy for different coefficient values. These results are included in a table below, and will also be added to the appendix.
>
> Coefficient values for optimal layer in gemma-2-9b-it (kind condition):
>
> | Steering coefficient | Accuracy |
> | :--- | :--- |
> | 1.0 | 0.72 |
> | 1.2 | 0.73 |
> | 1.4 | 0.73 |
> | 5.0 | 0.73 |
>
>
> Anonymized plots for selecting the best layer can be seen here, and will be added to the appendix.
>
> https://imgur.com/a/obC7Gqt

---

### Author Response · Authors · 2025-12-02
**Summary of Main Changes in Response to Reviews**

Firstly, we want to once again thank the reviewers who raised several useful points of feedback, many of which have inspired new analyses and changes, which we summarize below.
We appreciate the close reading of our manuscript and their noting of points that could use more clarification. We have included several new plots using anonymized links in the individual reviewer responses, and have noted details which we will include in both the appendix and the extra page, if the paper were to be accepted. We hope our changes have addressed the key concerns and limitations.


* **Expanded Model Scope & Generalization to larger models**: To address the core concern regarding how our results might not generalize to larger models, we have replicated key prompting experiments with a frontier model OpenAI o3. We believe this best balances the computational cost/infeasability of running the full suite of intervention analyses on >30B models, which we don’t have the infrastructure for, and generalizing our results to the largest and best performing reasoning models. Our analyses showed that despite high accuracy, the frontier model exhibits the same misalignment between 'size' and 'kind' representations observed in smaller models, suggesting our findings generalize to larger model sizes (R2 (vtyN)).

* **Enhanced Human Baselines & Robustness to order effects**: Thanks to the reviewers’ helpful suggestions to include more reliable human baselines and evaluate model response robustness we include 2 new analyses:
    * **Human Majority Vote**: We added a "human majority vote" baseline to Figure 3 to provide a more rigorous standard for accuracy than the previous embedding-based metric (while still including the embedding-based accuracy) (R1 (FSmf)).

    * **Order Effects**: We conducted a sensitivity analysis by inverting the order of options in the triplet task. We found high consistency in model choices (88.6%) and accuracy, ruling out ordering bias as a confound in our design (R2 (vtyN)).

* **Improved Statistical rigor in Reporting**: We overhaul the statistical reporting in Section 4 to provide the necessary depth and precision:
Error Estimation: We added permutation tests to estimate standard errors for our Procrustes $R^2$ values, adding error bars to Figure 3 to rule out spurious correlations (R3 (mvQ2)).

    * **Clarity in Regression Model reporting**: We detailed the logistic regression models used to predict accuracy and the Fisher r-to-z transforms used to compare alignment across conditions (R1 (FSmf)).

    * **Dimensionality & Hyperparameter Robustness**: We added analyses to justify our parameter choices:
* **Embedding Dimensionality Robustness**: We analyzed test loss across dimensions $d=2\dots6$. We confirmed that $d=2$ is sufficient for capturing representational structure, validating our choice to match human dimensionality (R3 (mvQ2)).

* **Steering Sensitivity**: Sensitivity analysis revealed that variability in the steering coefficient (values 1.0–5.0) and confirmed that our layer-selection approach offers a "best-case" estimate for each method, as intended, reinforcing the finding that even optimal steering yields poor alignment (R1 (FSmf)).

* **Methodological Clarifications**: We expanded the text to address specific implementation details:
   * **Parsing**: We clarified our token-matching strategy to ensure response  validity (R3 (mvQ2)).
   * **Task Validity**: We added citations (e.g., Muttenthaler et al., 2025) validating the triadic similarity task as a robust measure of deep semantic knowledge (R2 (vtyN)).

---

### Note · Program_Chairs · 2026-01-17
**Submission Desk Rejected by Program Chairs**

The following references in this submission do not refer to real documents and/or have major errors in bibliographic information:

 Jos De Bruin, Thomas Bourguignon, Mouhamadou Biran, Walid Saoud, Pierre MorizetMahoudeaux, Karine Tasso, Nicolas Chanez, Laurent Perrinet, Kathinka Evers, Claire Montfroy, et al. Strong and weak alignment of large language models with human values. Scientific Reports, 14(1):17428, 2024.